# Bayesian Additive Regression Trees for Exponential Family Distributions: A Theoretical Perspective

## Abstract

Bayesian Additive Regression Trees (BART) are a powerful ensemble learning technique for modeling nonlinear regression functions, which along with predictions also provide posterior uncertainty estimates unlike frequentist methods designed for similar tasks, such as random forests. Although initially BART was proposed for predicting only continuous and binary response variables, over the years multiple extensions have emerged that are suitable for estimating a wider class of response variables (e.g. categorical and count data) in a multitude of application areas. In this paper we describe a Generalized framework for Bayesian trees and their additive ensembles where the response variable comes from an exponential family distribution and hence encompasses a majority of these variants of BART. We derive sufficient conditions on the response distribution, under which the posterior concentrates at a near minimax rate. Our results provide theoretical justification for the empirical success of BART and its variants. In addition, the sufficient conditions provide important insights into practical model specifications such as the choice of link functions.

## 1 Introduction

Additive ensemble of Bayesian trees (Chipman et al., 1998; Denison et al., 1998), more popularly known as Bayesian additive regression trees (BART) (Chipman et al., 2010) is a flexible semiparametric tool that has been extremely successful in numerous high dimensional classification and regression tasks. Aided by efficient software implementations, (Sparapani et al. (2019), Bleich et al. (2014), Pratola et al. (2014) and He et al. (2019)), BART has thrived in a wide range of application areas, including causal inference (Hill, 2011; Hill & Su, 2013; Hahn et al., 2017), interaction detection (Du & Linero, 2019), survival analysis (Sparapani et al., 2016), time series analysis (Taddy et al., 2011; Deshpande et al., 2020) and variable selection (Bleich et al., 2014; Linero, 2018; Liu et al., 2018; Liu & Rockova, 2020), to name a few. Even though BART was initially proposed for predicting univariate continuous and binary response variables, due to its flexibility and impressive performance, multiple extensions have emerged that are suitable for a wide variety of both univariate and multivariate response variables (e.g. categorical and count data (Murray, 2021; **?**), heteroscedastic responses (Pratola et al., 2019; **?**)) and / or the target regression surface is of a constrained nature (e.g. monotone BART (Chipman et al., 2016), varying coefficient BART (Deshpande et al., 2020), BART with targeted smoothing (Starling et al., 2020) etc.).

Despite a long history of empirical success, theoretical studies on Bayesian trees and forests is a relatively new area of research. Recently emerging results along this line are geared towards providing a theoretical perspective on why these models have been so successful in a wide range of applications. Among the initial developments, Rockova & Saha (2019) and Rockova et al. (2020) demonstrated that the posterior concentration rate of BART equals to the minimax rate up to a logarithmic factor for various tree priors. Built on these findings, Rockova (2020) derived a semiparametric Bernstein von-Mises theorem for the BART estimator. Extensions of BART, adapted to various special function types have also been studied from a theoretical perspective: Linero & Yang (2017) studied a version of BART suitable for smooth function estimation; Castillo & Ro$v$cková (2021) conducted a multiscale analysis of BART and Jeong & Rockova (2020) derived posterior concentration results for anisotropic

functions. In this paper we study the posterior concentration rates of a generalized version of BART suitable, thereby supplementing this newly emerging area of research.

We formulate a Generalized BART (G-BART) model that extends the existing theoretical developments in several directions. Firstly while existing results focus on Gaussian response variables, we allow the response to come from an exponential family distribution. Hence G-BART can be regarded as semiparametric extensions of the widely popular 'Generalized Linear Models' (GLM) (Nelder & Wedderburn, 1972). Many prominent Bayesian CART and BART models used in practice (Denison et al., 1998; Chipman et al., 2010; Murray, 2021), including the traditional BART model (Chipman et al., 2010), can be viewed as a special case of this generalized extension. Therefore theoretical properties of these conventional adaptations of BART can be studied as direct corollaries of analogous properties for the G-BART model.

Secondly, existing results (Rockova et al., 2020; Rockova & Saha, 2019; Linero & Yang, 2017) assume that the underlying regression function is Hölder continuous. However, given the efficacy of BART models in a variety of complex applications, the assumption of Hölder continuity seems too restrictive. In this paper we demonstrate that similar posterior optimality results can be obtained for non-smooth functions as well, such as step functions and monotone functions, thus extending the theoretical findings on BART beyond the assumption of Hölder continuity.

Finally, the BART model Chipman et al. (2010) approximate the regression functions through step functions and assumes that these step heights come from a Gaussian distribution. All subsequent theoretical and empirical developments have adopted this specification. In the G-BART setup we assume that the distribution of these step heights belong to a broader family of distributions that include both the Gaussian distribution and also some thicker tailed distributions like Laplace. We demonstrate that the BART model maintains a near-minimax posterior concentration rate, if the step heights come from any of the distributions belonging to this broader family, thus providing a wide range of distributional choices without sacrificing fast posterior concentration.

The paper is organized as follows: Section 2 describes the generalized BART model, Section 3 discusses the notion of posterior concentration, followed by the main theoretical results in Section 4. Broader implications of these results are described in Section 5. Finally, Section 6 concludes with a discussion. Proofs of the main results are provided in the supplementary material.

## 1.1 OUR CONTRIBUTIONS

To summarize our previous discussion, we now briefly highlight our key contributions.

**Response distribution:** Existing theoretical results on BART focus on univariate Gaussian response variables. In contrast, we assume that the response variable comes from a multivariate exponential family distribution and derive sufficient conditions on the response density under which the posterior concentrates at a near-minimax rate.

**Step Size distribution:** Instead of assigning a Gaussian distribution on the step-heights associated to the BART model, we impose sufficient conditions on the cumulative distribution function that guarantee a near-optimal posterior concentration rate. The resulting family of distributions encompasses the Gaussian distribution along with several thicker tailed distributions like Laplace, which is suitable for modeling heterogeneous populations, thus widening modeling choices for empirical applications.

**Types of functions:** The objective of BART models is to estimate unknown functions that characterize the relationship between the response and the covariates. Existing results on BART assume this function to be Hölder continuous. We extend these results for non-continuous function spaces such as monotone functions and step functions supported on an axes-paralleled partition. The results on step functions in conjuction with the "simple function approximation theorem" (Stein & Shakarchi, 2009), can be useful for deriving posterior concentration rates for more general class of functions.

In addition to the above, we also prove that specific model choices such as the choice of link functions can influence the posterior concentration rate of the G-BART model. Thus the results discussed in this paper can also provide useful insights into selecting link functions that provide faster concentration rates of the posterior, possibly leading to better empirical performance.

## 1.2 NOTATIONS:

For any two real numbers $a$ and $b$, $a \vee b$ will denote the maximum of $a$ and $b$. The notations $\gtrsim$ and $\lesssim$ will stand for "greater than or equal to up to a constant" and "less than or equal to up to a constant", respectively. The symbol $P_f$ will abbreviate $\int f dP$ and $\mathbb{P}_f^{(n)} = \prod_{i=1}^n \mathbb{P}_f^i$ will denote the $n$-fold product measure of the $n$ independent observations, where the $i$-th observation comes from the distribution $P_f^i$. Let $h(f,g) = \left(\int (\sqrt{f} - \sqrt{g})^2 d\mu\right)^{1/2}$ and $K(f,g) = \int f \log(f/g) d\mu$ denote the Hellinger distance and the Kullback-Leibler divergence, respectively between any two non-negative densities $f$ and $g$ with respect to a measure $\mu$. We define another discrepancy measure $V(f,g) = \int f \left(\log(f/g)\right)^2 d\mu$. Finally, for any set of real vectors $\boldsymbol{X}_1, \dots, \boldsymbol{X}_n \in \mathbb{R}^q$ of size $n$, define the average discrepancy measures $H_n(f,g) = \frac{1}{n}\sum_{i=1}^n H\left(f(\boldsymbol{X}_i), g(\boldsymbol{X}_i)\right)$, $K_n(f,g) = \frac{1}{n}\sum_{i=1}^n K\left(f(\boldsymbol{X}_i), g(\boldsymbol{X}_i)\right)$ and $V_n(f,g) = \frac{1}{n}\sum_{i=1}^n V\left(f(\boldsymbol{X}_i), g(\boldsymbol{X}_i)\right)$, where $f(\theta)$ and $g(\theta)$ denote the densities $f$ and $g$ with respect to parameter $\theta$. Also, for any $L_p$ norm $\|\cdot\|_p$, define the average norm $\|f - g\|_{p,n} = \frac{1}{n}\sum_{i=1}^n \|f - g\|_p$.

## 2 THE GENERALIZED BART PRIOR

The BART method of Chipman et al. (2010) is a prominent example of Bayesian ensemble learning, where individual shallow trees are entwined together into a forest, that is capable of estimating a wide variety of nonlinear functions with exceptional accuracy, while simultaneously accounting for different orders of interactions among the covariates. Building upon BART, we describe a generalized model, where the response variable is assumed to come from an exponential family distribution. For continuous Gaussian response variables, this generalized BART model reduces to the original BART prior of (Chipman et al., 2010).

The data setup under consideration consists of $\boldsymbol{Y}_i = (y_{i1}, \dots, y_{ip})' \in \mathbb{R}^p$, a set of $p$-dimensional outputs, and $\boldsymbol{X}_i = (x_{i1}, \dots, x_{iq})' \in [0,1]^q$, a set of $q$ dimensional inputs for $1 \le i \le n$. We assume $\boldsymbol{Y}$ follows some distribution in the exponential family with density of the following form:

$$P_{f_0}(\boldsymbol{Y} \mid \boldsymbol{X}) = h(\boldsymbol{Y})g\left[f_0(\boldsymbol{X})\right]\exp\left[\eta\left(f_0(\boldsymbol{X})\right)^T T(\boldsymbol{Y})\right], \tag{1}$$

where $h : \mathbb{R}^p \to \mathbb{R}$, $g : \mathbb{R} \to \mathbb{R}$, $\eta : \mathbb{R}^p \to \mathbb{R}^J$, $T : \mathbb{R}^p \to \mathbb{R}^J$ for some integer $J$ and $f_0 : \mathbb{R}^q \to \mathbb{R}^D$, for some integer $D$, are all real valued functions. Among these functions, $h$, $g$, $\eta$ and $T$ are usually *known* depending on the nature of the response $\boldsymbol{Y}$. The function $f_0$, connecting the input $\boldsymbol{X}$ with the output $\boldsymbol{Y}$, is the only unknown function and estimating this function is the primary objective of the G-BART estimator.

We assume that $f_0$ is an unconstrained function, i.e. the range of $f_0$ is the entire space $\mathbb{R}^D$ for some integer $D$. A suitable link function $\Psi(\cdot)$ is used to transform $f_0$ to the natural parameter of the distribution of $\boldsymbol{Y}$, which is often constrained. For example, for the binary classification problem, $\boldsymbol{Y} \sim Bernoulli\left(p(\boldsymbol{X})\right)$. Here the natural parameter $p(\boldsymbol{X}) \in (0,1)$ is restricted and hence we can use $\Psi(z) = \frac{1}{1+\exp(-z)}$, the logistic function (or a probit function, as in Chipman et al. (2010)) to map the unconstrained function $f_0(\boldsymbol{X})$ to the natural parameter $p(\boldsymbol{X})$. There are usually several different choices for the link function. As we will see in Section 5, the BART estimator might have different posterior concentration rates depending on which link function is used to transform the function $f_0$ to the natural parameter of the response distribution.

The univariate regression and the two-class classification problem considered in the original BART paper (Chipman et al., 2010) and many of its important extensions, such as the multi-class classification and the log-linear BART (Murray, 2021) for categorical and count responses can be formulated as special cases of equation 1. The specific forms of the functions $h, g, \eta$ and $T$ for continuous regression and multi-class classification are given in Table 1.

Next a regression tree is used to reconstruct the unknown function $f_0 : \mathbb{R}^q \to \mathbb{R}^D$ via a mapping $f_{\boldsymbol{T},\boldsymbol{\beta}} : [0,1]^q \to \mathbb{R}^D$ so that $f_{\boldsymbol{T},\boldsymbol{\beta}}(\boldsymbol{X}) \approx f_0(\boldsymbol{X})$ for $\boldsymbol{X} \notin \{\boldsymbol{X}_i\}_{i=1}^n$. Each such mapping is essentially a step function of the form

$$f_{\boldsymbol{T},\boldsymbol{\beta}}(\boldsymbol{X}) = \sum_{k=1}^K \beta_k \mathbb{I}(\boldsymbol{X} \in \Omega_k) \tag{2}$$

Table 1: Univariate Regression (column 2) and Multi-class Classification (column 3), as special cases of the Generalized BART model. $\Phi$ denotes the $Softmax$ function and $\mathcal{M}(\cdot)$ denotes the $Multinomial(1; \cdot)$ distribution. $(\{\mathbb{I}\{Y = i\}\}_{i=1}^p)'$ denotes the row vector where the $i$-th coordinate equals to one if $\boldsymbol{Y}$ belongs to class $i$ and zero otherwise.

| Response ($\boldsymbol{Y}$) | Continuous | Categorical |
|---|---|---|
| Dist.($\boldsymbol{Y}$) | $\mathcal{N}\left(f_0(\boldsymbol{X}), \sigma^2\right)$ | $\mathcal{M}\left(\Phi(f_0(\boldsymbol{X}))\right)$ |
| $h(\boldsymbol{Y})$ | $1/\sqrt{2\pi}\sigma$ | 1 |
| $g\left(f_0(\boldsymbol{X})\right)$ | $\exp\left(-f_0(\boldsymbol{X})^2/\sigma^2\right)$ | 1 |
| $\eta\left(f_0(\boldsymbol{X})\right)$ | $(f_0(\boldsymbol{X}), 1)$ | $f_0(\boldsymbol{X})$ |
| $T(\boldsymbol{Y})$ | $\left(2Y/\sigma^2, -Y^2/\sigma^2\right)$ | $(\{\mathbb{I}\{Y = i\}\}_{i=1}^p)'$ |
| $f_0(\boldsymbol{X})$ | $\mathbb{R}^q \to \mathbb{R}$ | $\mathbb{R}^q \to \mathbb{R}^{p-1}$ |

supported on a tree-shaped partition $\boldsymbol{T} = \{\Omega_k\}_{k=1}^K$ and specified by a vector of step heights $\boldsymbol{\beta} = (\beta_1, \ldots, \beta_K)'$. The vector $\beta_k \in \mathbb{R}^p$ represents the value of the expected response inside the $k$-th cell of the partition $\Omega_k$.

Bayesian additive trees consist of an ensemble of multiple shallow trees, each of which is intended to be a weak learner, geared towards addressing a slightly different aspect of the prediction problem. These trees are then woven into an *additive* forest mapping of the form

$$f_{\boldsymbol{E}, \boldsymbol{B}}(\boldsymbol{x}) = \sum_{t=1}^T f_{\boldsymbol{T}_t, \boldsymbol{\beta}_t}(\boldsymbol{x}), \tag{3}$$

where each $f_{\boldsymbol{T}_t, \boldsymbol{\beta}_t}(\boldsymbol{x})$ is of the form equation 2, $\boldsymbol{E} = \{\boldsymbol{T}_1, \ldots, \boldsymbol{T}_T\}$ is an ensemble of $T$ trees and $\boldsymbol{B} = \{\boldsymbol{\beta}_1, \ldots, \boldsymbol{\beta}_T\}'$ is a collection of jump sizes corresponding to the $T$ trees.

Since each individual member of the approximating space is a step function of the form equation 3, supported on a Bayesian additive forest, the prior distribution should include three components: (i) a prior $\pi(T)$ on the number of trees $T$ in the ensemble, (ii) a prior on individual tree partitions $\pi(\boldsymbol{T})$ and their collaboration within the ensemble and (iii) given a single tree partition $\boldsymbol{T}$, a prior $\pi(\boldsymbol{\beta} \,|\, \boldsymbol{T})$ has to be imposed on the individual step heights $\boldsymbol{\beta}$.

In this paper we follow the recommendation by Chipman et al. (2010) and assume the number of trees $T$ to be fixed at a large value (e.g. $T = 200$ for regression and $T = 50$ for classification). Alternatively, one can also assign a prior with higher dispersion, as in Rockova et al. (2020) and Linero & Yang (2017) and replicate the steps of the proofs provided in the appendix with minor modifications.

Given the total number of trees in the ensemble, individual trees are assumed to be independent and identically distributed with some distribution $\pi(\boldsymbol{T})$. This reduces the prior on the ensemble to be of the form

$$\pi(\boldsymbol{E}, \boldsymbol{B}) = \prod_{t=1}^T \pi(\boldsymbol{T}_t)\pi(\boldsymbol{\beta}_t \,|\, \boldsymbol{T}_t), \tag{4}$$

where $\pi(\boldsymbol{T}_t)$ is the prior probability of a partition $\boldsymbol{T}_t$, while $\pi(\boldsymbol{\beta}_t \,|\, \boldsymbol{T}_t)$ is the prior distribution over the jump sizes. The specific forms of the priors $\pi(\boldsymbol{T})$ and $\pi(\boldsymbol{\beta} \,|\, \boldsymbol{T})$ are described below.

### 2.1 Prior on partitions

We consider two distinct prior distributions on the partitions $\pi(\boldsymbol{T})$ proposed by Chipman et al. (1998) and Denison et al. (1998) respectively. The posterior concentration results discussed in Section 4 are applicable to both these priors. Chipman et al. (1998) specifies the prior over trees implicitly as a tree generating stochastic process, described as follows:

1. Start with a single leaf (a root node) encompassing the entire covariate space.

2. Split a terminal node, say $\Omega$, with a probability

$$p_{split}(\Omega) \propto \alpha^{-d(\Omega)} \text{ for some } 0 < \alpha < 1/2. \tag{5}$$

where $d(\Omega)$ is the depth of the node $\Omega$ in the tree architecture. This choice, motivated by (Rockova & Saha, 2019) (see section **??**).

3. If the node $\Omega$ splits, assign a splitting rule and create left and right children nodes. The splitting rule consists of picking a split variable $j$ uniformly from available directions $\{1, \ldots, p\}$ and picking a split point $c$ uniformly from available data values $x_{1j}, \ldots, x_{nj}$.

A description of the prior proposed by Denison et al. (1998) is given in Section A.1 in the supplementary material.

## 2.2 **Prior on step heights**

We impose a broad class of priors on the step heights that incorporate the corresponding component of the classical BART model as a special case. Given a tree partition $\boldsymbol{T}_t$ with $K_t$ steps, Chipman et al. (2010) considers identically distributed independent Gaussian jumps with mean 0 and variance $\sigma^2$. In the G-BART set-up we assume that the $j$-th step height of the $t$-th tree, $\beta_{tj} \overset{i.i.d}{\sim} F_\beta$, where $F_\beta$ is any general distribution with the following property: for some constants $C_1, C_2, C_3$ such that $C_1 > 0$, $0 < C_2 \leq 2$ and $C_3 > 0$,

$$F_\beta(\|\beta\|_\infty \leq t) \gtrsim \left( e^{-C_1 t^{C_2}} t \right)^p \quad \text{for } 0 < t \leq 1 \tag{6}$$

and

$$F_\beta(\|\beta\|_\infty \geq t) \lesssim e^{-C_3 t} \quad \text{for } t \geq 1 \tag{7}$$

where $\|\cdot\|_\infty$ represents the $L_\infty$ norm and $F_\beta(\|\beta\|_\infty \geq t)$ denotes the tail probability of the distribution on the step heights $\beta \in \mathbb{R}^p$. Both the multivariate Gaussian and the multivariate Laplace distribution come from this family of distributions and so do any sub-Gaussian distributions. A proof of these statements is provided in the appendix. We will see in Section 4.1 and Section 4.3 that these conditions are *sufficient* to guarantee that the G-BART estimator has a near-optimal posterior concentration rate.

However we should note that the conditions equation 6-equation 7, although *sufficient*, are not *necessary* conditions and distributional assumptions on the step sizes that do not satisfy these conditions, might still guarantee a near-optimal posterior concentration rate. For such an example, please refer to the 'classification with Dirichlet step-heights' in the supplementary material.

## 3 POSTERIOR CONCENTRATION

Posterior concentration statements are a prominent artifact in Bayesian nonparametrics, where the primary motivation is to examine the quality of a Bayesian procedure, by studying the learning rate of its posterior, i.e. the rate at which the posterior distribution, centralizes around the truth as the sample size $n \to \infty$. In empirical settings, posterior concentration results have often influenced the proposal and fine-tuning of priors. Oftentimes seemingly unremarkable priors give rise to capricious outcomes, specially in the infinite-dimensional parameter spaces, such as the one considered here (Cox (1993), Diaconis & Freedman (1986)) and designing well-behaved priors turn out to be of utmost importance, thus further reinstating the importance of posterior concentration statements.

The Bayesian approach proceeds by imposing a prior measure $\Pi(\cdot)$ on $\mathcal{F}$, the set of all estimators of $f_0$. For the G-BART models this corresponds to the set of all step functions supported on an additive ensemble of Bayesian trees. Given observed data $\boldsymbol{Y}^{(n)} = (Y_1, \ldots, Y_n)'$, the inference about $f_0$ is solely dependent on the posterior distribution

$$\Pi(A \mid \boldsymbol{Y}^{(n)}) = \frac{\int_A \prod_{i=1}^n \Pi_f(Y_i \mid \boldsymbol{X}_i) \mathrm{d}\,\Pi(f)}{\int \prod_{i=1}^n \Pi_f(Y_i \mid \boldsymbol{X}_i) \mathrm{d}\,\Pi(f)} \quad \forall A \in \mathcal{B}$$

where $\mathcal{B}$ is a $\sigma$-field on $\mathcal{F}$ and where $\Pi_f(Y_i \mid \boldsymbol{X}_i)$ is the conditional likelihood function for the output $Y_i$, given the covariates $\boldsymbol{X}_i$, under the parameterization $f$.

Ideally under a suitable prior, the posterior should put most of its probability mass around a small neighborhood of the true function and as the sample size increases, the diameter of this neighborhood should go to zero at a fast pace. Formally speaking, for a given sample size $n$, if we examine an $\varepsilon_n$-neighborhood of the true function $\mathcal{A}_{\varepsilon_n}$, for some $\varepsilon_n \to 0$ and $n\varepsilon_n^2 \to \infty$, we should expect

$$\Pi(\mathcal{A}_{\varepsilon_n}^c \mid \boldsymbol{Y}^{(n)}) \to 0 \quad \text{in } \mathbb{P}_{f_0}^{(n)}\text{-probability as } n \to \infty, \tag{8}$$

where $\mathcal{A}_{\varepsilon_n}^c$ denotes the complement of the neighborhood $\mathcal{A}_{\varepsilon_n}$.

In the context of G-BART, given observed data $\boldsymbol{Y}^{(n)} = (\boldsymbol{Y}_1, \ldots, \boldsymbol{Y}_n)'$, we are interested in evaluating whether the posterior concentrates around the true likelihood $\mathbb{P}_{f_0}^{(n)} = \prod_{i=1}^n P_{f_0}^i$ at a near-minimax rate, where $P_{f_0}^i = P_{f_0}(\boldsymbol{Y}_i \mid \boldsymbol{X}_i)$ is of the form equation 1, for $i = 1, \ldots, n$. Following the suggestions of Ghosal et al. (2007), we look at the smallest $H_n$-neighborhoods around $\mathbb{P}_{f_0}^{(n)}$ that contain the bulk of the posterior probability. Specifically, for a diameter $\varepsilon > 0$ define

$$\mathcal{A}_\varepsilon = \{f \in \mathcal{F} : H_n(P_f, P_{f_0}) \leq \varepsilon\} \tag{9}$$

Theorem 4 of Ghosal et al. (2007) demonstrates that the statement equation 8 can be proved by verifying three sufficient conditions. The first condition, henceforth referred to as the "entropy condition" specifies that

$$\sup_{\varepsilon > \varepsilon_n} \log N\left(\tfrac{\varepsilon}{36}; \mathbb{F}_n \cap \mathcal{A}_\varepsilon; H_n\right) \lesssim n\,\varepsilon_n^2, \tag{C1}$$

where $N(\varepsilon; \Omega; d)$ denotes the $\varepsilon$-covering number of a set $\Omega$ for a semimetric $d$, i.e. the minimal number of $d$-balls of radius $\varepsilon$ needed to cover the set $\Omega$ and $\{\mathbb{F}_n\}_{n \geq 1}$ denotes an increasing sequence of approximating sieves. The sequence of sieves used in this paper is described in the appendix.

The second condition requires that the prior puts enough mass around the true likelihood $\mathbb{P}_{f_0}^{(n)}$, meaning that for a given sample size $n \in \mathbb{N} \setminus \{0\}$ and for some $d > 2$,

$$\Pi(f \in \mathcal{F} : K_n(f, f_0) \vee V_n(f, f_0) \leq \varepsilon_n^2) \gtrsim e^{-d\,n\,\varepsilon_n^2}, \tag{C2}$$

where $K_n$ and $V_n$ are the Kullback-Leibler divergence and the variation, averaged over the observed data points.

The final condition, referred to as the "prior decay rate condition" stipulates that the sequence of sieves $\mathbb{F}_n \uparrow \mathcal{F}$ captures the entire parameter space with increasing accuracy, in the sense that the complementary space $\mathcal{F} \backslash \mathbb{F}_n$ has negligible prior probability mass for large values of $n$.

$$\Pi(\mathcal{F} \backslash \mathbb{F}_n) = o(e^{-(d+2)\,n\,\varepsilon_n^2}) \tag{C3}$$

The results of type equation 8 quantify not only the typical distance between a point estimator (posterior mean/median) and the truth, but also the typical spread of the posterior around the truth and hence are stronger than 'posterior consistency' statements. These results pave the way for further uncertainty quantification statements such as semiparametric Bernstein-von Mises theorem (Castillo et al., 2014).

# 4 MAIN RESULTS

In this section we describe our main theoretical findings, which describe the posterior concentration rates of the generalized Bayesian trees and their additive ensembles (G-BART), when the true function $f_0$ connecting the response $\boldsymbol{Y}$ with the covariates $\boldsymbol{X}$, is either (a) a step function (Theorem 4.1), or (b) a monotone function (Theorem 4.3), or (c) a $\nu$-Hölder continuous function with $0 < \nu \leq 1$ (Theorem 4.4). We make two important assumptions:

**Assumption 1:** Let $\boldsymbol{Y}_1, \ldots, \boldsymbol{Y}_n \sim P_f$, where $P_f$ denotes a probability density function of the form equation 1, such that, $\eta(z) = z$ and there exists strictly increasing positive sequences $\{C_g^n\}_{n \geq 1}$ and $\{C_\beta^n\}_{n \geq 1}$, such that

$$\left| \frac{\nabla g(\boldsymbol{\beta})}{g(\boldsymbol{\beta})} \right| \leq C_g^n \mathbf{1}_p, \quad \forall \boldsymbol{\beta} \in B_n = \left\{\boldsymbol{\beta} : \|\boldsymbol{\beta}\|_\infty \leq C_\beta^n\right\}, \tag{10}$$

where $\mathbf{1}_p = (1, \ldots, 1) \in \mathbb{R}^p$ denotes a $p$-dimensional vector of ones and $\nabla g$ denotes the vector of partial derivatives. We assume $\{C_g^n\} \vee \{C_\beta^n\} \lesssim n^M$ for some $M > 0$. The significance is that the function $g(\cdot)$ should not change too rapidly, and the higher the sample size the larger the rate of change is allowed. The above assumption is satisfied by most distributions commonly used in the regression and classification settings, as will be demonstrated in Section 5.

**Assumption 2:** For a k-d tree partition, $\widehat{\mathcal{T}} = \{\widehat{\Omega_k}\}$, with $K = 2^{ps}$-many leaves, the dataset $\{\boldsymbol{X}_1, \ldots, \boldsymbol{X}_n\}$ satisfies the following condition: for any nonnegative integer $s$, there exists some large enough constant $M > 0$ such that

$$\max_{1 \leq k \leq K} \text{diam}(\widehat{\Omega_k}) < M \sum_{k=1}^{K} \mu(\Omega_k) \text{diam}(\widehat{\Omega_k}), \tag{11}$$

where $\mu(\Omega_k) = \frac{1}{n} \sum_{i=1}^{n} \mathbb{I}\{\boldsymbol{X}_i \in \Omega_k\}$ denotes the proportion of observations in the cell $\Omega_k$ and $\text{diam}(\Omega_k) = \max_{\boldsymbol{x}, \boldsymbol{y} \in \Omega_k} \|\boldsymbol{x} - \boldsymbol{y}\|_2$ denotes the spread of the cell $\Omega_k$ with respect to the $L_2$-norm.

## 4.1 RESULTS ON STEP-FUNCTIONS

Let us suppose $f_0$ is a step function supported on an axes-paralleled partition $\{\Omega_k\}_{k=1}^{K_0}$. For any such step function $f_0$, we define the *complexity of $f_0$*, as the smallest $K$ such that there exists a partition $\{\Omega_k\}_{k=1}^{K}$ with $K$ cells, for which the step function $f(x) = \sum_{k=1}^{K} \beta_k \mathbb{I}\{x \in \Omega_k\}$ can approximate $f_0$ without any error, for some step heights $(\beta_1, \ldots, \beta_K) \in \mathbb{R}^K$. This complexity number, denoted by $K_{f_0}$, depends on the true number of step $K_0$, the diameter of the intervals $\{\Omega_k\}_{k=1}^{K_0}$, and the number of covariates $q$. The actual minimax rate for approximating such piecewise-constant functions $f_0$ with $K_0 > 2$ pieces, is $n^{-1/2} \sqrt{K_0 \log(n/K_0)}$ (Gao et al., 2017). The following theorem shows that the posterior concentration rate of G-BART is almost equal to the minimax rate, except that $K_0$ gets replaced by $K_{f_0}$. The discrepancy is an unavoidable consequence of the fact that the true number of steps $K_0$ is unknown. Had this information been available, the G-BART estimator would have attained the exact minimax rate.

**Theorem 4.1.** *If we assume that the distribution of the step-sizes satisfies equation 6 and equation 7, then under Assumptions 1 and 2 with $q \lesssim \sqrt{\log n}$, the generalized BART estimator satisfies the following property:*

*If $f_0$ is a step-function, supported on an axes-paralleled partition, with complexity $K_{f_0} \lesssim \sqrt{n}$ and $\|f_0\|_\infty \lesssim \sqrt{\log n}$, then with $\varepsilon_n = n^{-1/2} \sqrt{K_{f_0} \log^{2\gamma}(n/K_{f_0})}$ and $\gamma > 1/2$,*

$$\Pi\left(f \in \mathcal{F} : H_n(\mathbb{P}_f, \mathbb{P}_{f_0}) > \varepsilon_n \mid \boldsymbol{Y}^{(n)}\right) \to 0,$$

*in $\mathbb{P}_{f_0}^{(n)}$-probability, as $n, q \to \infty$.*

## 4.2 RESULTS ON MONOTONE FUNCTIONS

An important implication of Theorem 4.1 is that posterior concentration results on step functions can potentially build the foundation for similar results on broader class of functions, aided by the "simple function approximation theorem" (Stein & Shakarchi, 2009), which states that for any measurable function $f$ on $\mathcal{E} \subseteq \mathbb{R}^q$, there exists a sequence of step functions $\{f_k\}$ which converges point-wise to $f$ almost everywhere (Stein & Shakarchi, 2009). As a corollary to this theorem, we can derive the following result on the set of all monotone functions. A function $f_0 : \mathbb{R}^q \to \mathbb{R}$ is defined as monotone increasing (or decreasing) if $f_0(\boldsymbol{x}_1) \geq f_0(\boldsymbol{x}_2)$ (or, $f_0(\boldsymbol{x}_1) \leq f_0(\boldsymbol{x}_2)$) for all $\boldsymbol{x}_1, \boldsymbol{x}_2$ such that every coordinate of $\boldsymbol{x}_1$ is greater than or equal to the corresponding coordinate of $\boldsymbol{x}_2$.

**Lemma 4.2.** *Any **monotone** bounded function $f_0$ can be approximated with arbitrary precision $\varepsilon$, by a step function supported on a k-d tree partition with number of leaves $K_{f_0}(\varepsilon) \geq \lceil 1/\varepsilon \rceil$. We define $K_{f_0}(\varepsilon)$ to be the complexity of the monotone function $f_0$ with respect to $\varepsilon > 0$.*

The complexity $K_{f_0}(\varepsilon)$ also depends on the dimension of the domain $q$ as well as on the magnitude of the true function $\|f_0\|_\infty$. This paves the way for deriving the posterior concentration rate of G-BART when the true function $f_0(\cdot)$ connecting the covariates $\boldsymbol{X}$ with a univariate response $\boldsymbol{Y}$ is a monotone function. The minimax rate of estimation for such densities is $n^{-1/(2+q)}$ (Biau & Devroye, 2003). The following theorem states that the posterior concentration rate of G-BART equals to this optimum rate up to a logarithmic function, provided that the magnitude of the true function $f_0$ is not "too large".

**Theorem 4.3.** *If the distribution of the step-sizes satisfies equation 6 and equation 7, then under Assumptions 1 and 2 with $q \lesssim \sqrt{\log n}$, the generalized BART estimator satisfies the following property:*

*If the true function $f_0 : \mathbb{R}^q \to \mathbb{R}$ is monotonic on every coordinate, with $\|f_0\|_\infty \lesssim \sqrt{\log n}$, then with $\varepsilon_n = n^{-1/(2+q)}\sqrt{\log n}$,*

$$\Pi\left(f \in \mathcal{F} : H_n(\mathbb{P}_f, \mathbb{P}_{f_0}) > \varepsilon_n \mid \boldsymbol{Y}^{(n)}\right) \to 0,$$

*in $\mathbb{P}_{f_0}^{(n)}$-probability, as $n, q \to \infty$.*

The above result demonstrates that the Generalized BART model adapts to monotonic patterns in the true function $f_0$, without any additional prior assumptions.

### 4.3 RESULTS ON HÖLDER CONTINUOUS FUNCTIONS

This section describes the posterior concentration results on G-BART when the true function $f_0$ connecting $\boldsymbol{X}$ with $\boldsymbol{Y}$ is a $\nu$-Hölder continuous function with $0 < \nu \leq 1$. Rockova et al. (2020) and Rockova & Saha (2019) proved that the posterior concentration rates of the BART model (under the priors of Denison et al. (1998) and Chipman et al. (2010) respectively) are equal to $n^{-\alpha/(2\alpha+q)}$, the minimax rate of estimation for such functions (Stone, 1982), except for a logarithmic factor. These results can be derived as direct corollaries of the following theorem for G-BART, when $\boldsymbol{Y}$ is a univariate continuous response and the step-sizes are assumed to follow a Gaussian distribution.

**Theorem 4.4.** *If we assume that the distribution of the step-sizes satisfies equation 6 and equation 7, then under Assumptions 1 and 2 with $q \lesssim \sqrt{\log n}$, the generalized BART estimator satisfies the following property:*

*If $f_0$ is a $\nu$-Hölder continuous function with $0 < \nu \leq 1$, where $\|f_0\|_\infty \lesssim \sqrt{\log n}$, then with $\varepsilon_n = n^{-\alpha/(2\alpha+q)}\sqrt{\log n}$,*

$$\Pi\left(f \in \mathcal{F} : H_n(\mathbb{P}_f, \mathbb{P}_{f_0}) > \varepsilon_n \mid \boldsymbol{Y}^{(n)}\right) \to 0,$$

*in $\mathbb{P}_{f_0}^{(n)}$-probability, as $n, q \to \infty$.*

Interestingly, the posterior concentration rates derived in Theorems 4.1-4.4, do not depend on the number of trees $T$ in the generalized BART ensemble. In other words the concentration rate is equally valid for a single tree (i.e. $T = 1$), as well as for tree ensembles (i.e. $T > 1$), when the true regression function $f_0$ is $\nu$-Hölder continuous with $0 < \nu \leq 1$. However as has been seen in multiple empirical applications (Chipman et al., 2010), Bayesian forests consisting of multiple trees provide superior out-of-sample predictive performance, compared to a single tree, the reason being that multiple weak tree learners, when woven together into a forest, can accommodate a wider class of partitions, as opposed to a single tree. This can be reinforced by theoretical results, such as Theorem 6.1 of Rockova et al. (2020). When the true function $f_0$ is of the form $f_0 = \sum_{t=1}^{T_0} f_0^t$, where $f_0^t$ is a $\nu_t$-Hölder continuous function, with $0 \leq \nu^t \leq 1$, a forest with multiple trees have a posterior concentration rate equal to $\varepsilon_n^2 = \sum_{t=1}^{T_0} n^{-2\nu_t/(2\nu_t+p)} \log n$, provided $T_0 \lesssim n$, whereas single regression trees fail to recognize the additive nature of the true function and attain a slower concentration rate. A similar result is presented in Theorem 4 of Linero & Yang (2017), under a kernel-smoothed version of the BART prior.

## 5 IMPLICATIONS

The primary significance of Theorems 4.1, 4.3 and 4.4 is that these results provide a frequentist theoretical justification for superior empirical performance of generalized Bayesian trees and forests, claiming that the posterior concentrates around the truth at a near-optimal learning rate. As demonstrated below, we can show that the original BART model (Chipman et al., 2010), along with some of its commonly used variants (such as BART for multi-class classification and regression on count data) have near-optimal posterior concentration rates, as direct corollaries of Theorems 4.1 - 4.4. Another consequence of these results is that (see Section A.5 of the supplementary material), they show that the posterior distribution on the number of leaves in a generalized Bayesian tree does not exceed the optimal number of splits by more than a constant multiple and hence are resilient to overfitting.

**Continuous Regression:** For a (multivariate) continuous regression, assume that the response $\boldsymbol{Y} \mid \boldsymbol{X} \sim \mathcal{N}_p(\boldsymbol{\mu}(\boldsymbol{X}), \Sigma)$, for some positive definite $\Sigma$. The function $g(f_0(\boldsymbol{X})) = g(\boldsymbol{\mu}) =$

$e^{-\boldsymbol{\mu}^T \Sigma^{-1} \boldsymbol{\mu}/2}$ satisfies equation 10 with $B_n = [-n, n]^p$ and $C_g^n = n\lambda(\Sigma)$, where $\lambda(\Sigma)$ denotes the maximum eigenvalue of $\Sigma$. Hence from Theorems 4.1, 4.3 and 4.4, we can conclude that for continuous regression, the G-BART estimator has a near-minimax posterior concentration rate, provided that the true function $f_0$ connecting the input $\boldsymbol{X}$ with the output $\boldsymbol{Y}$ is either a step function, a monotone function or a $\nu$-Hölder continuous function with $0 < \nu \le 1$.

**Classification with Gaussian Step Heights:** For a $p$-class classification the response $\boldsymbol{Y}$ can be written as a $p$ dimensional binary vector that has 1 at the $l$-th coordinate if $\boldsymbol{Y}$ belongs to category $l \in \{1, \ldots, p\}$ and 0 elsewhere. We can assume $\boldsymbol{Y} \mid \boldsymbol{X} \sim \text{Multinomial}(1; \boldsymbol{\pi}(\boldsymbol{X}))$ for some $\boldsymbol{\pi} : \mathbb{R}^q \in (0, 1)^p$ such that $\boldsymbol{\pi}' \boldsymbol{1}_p = 1$. The unrestricted function $f_0(\boldsymbol{X})$ can be transformed to the natural parameter $\pi(\boldsymbol{X})$ by a logistic (softmax) or an inverse-probit link function (Chipman et al., 2010) denoted by $\Psi(\cdot)$, so that $\pi(\boldsymbol{X}) = \Psi(f_0(\boldsymbol{X}))$. In either case, the function $g(f_0(\boldsymbol{X})) = 1$ trivially satisfies condition equation 10. Hence from Theorem 4.1 and Theorem 4.4, we can conclude that the BART model for multi-class classification has a near-minimax posterior concentration rate.

For the same multi-class classification problem with $p$ classes described above, an alternative prior specification is recommended by Denison et al. (1998). Althogh this prior violates condition equation 6, we can show that this estimator has a near-optimal posterior concentration rate (proof in supplementary material). This demonstrates that the assumptions we make in Section 4 are merely *sufficient* but not *necessary* conditions for proving that the generalized Bayesian tree estimator has a near-minimax posterior concentration rate.

**Count Regression:** For count response variable, $\boldsymbol{Y} \sim Poisson\left[\lambda(\boldsymbol{X})\right]$ with $\lambda(\boldsymbol{X}) > 0$. There are several choices for the link function $\Psi(\cdot)$ to map the unconstrained function $f_0(\boldsymbol{X})$ to the constrained parameter $\lambda(\boldsymbol{X})$. The posterior concentration rate of the Generalized Bayesian tree estimator might differ depending on which link function is used. For example, if we use $\Psi(z) = \log\left(1 + \exp(z)\right)$, the softplus link function, then $g(f_0(\boldsymbol{X})) = 1/(1 + \exp\left(f_0(\boldsymbol{X})\right)$, trivially satisfies condition equation 10 and we can conclude that the generalized tree estimator has a near-minimax concentration rate from Theorems 4.1, 4.3 and 4.4. In contrast, if we use $\Psi(z) = \exp(z)$ as the link function, then $g(f_0(\boldsymbol{X})) = \exp\left(-\exp(f_0(\boldsymbol{X}))\right)$ does not satisfy the condition equation 10, when the true function $f_0$ is a $\nu$-Hölder continuous function. Therefore we cannot apply Theorem 4.4 anymore to imply that the generalized tree estimator has a near-optimal rate of posterior concentration. When $f_0$ is a step function with complexity $K_{f_0}$, the condition equation 10 is satisfied with $B_n = [-K_{f_0} \log n, K_{f_0} \log n]$ and $C_g^n = n^{K_{f_0}}$. The posterior concentration rate becomes $\varepsilon_n = n^{-\frac{1-\alpha}{2}} \sqrt{K_{f_0} \log^{2\eta}(n/K_{f_0})}$ under the assumption $K_{f_0} \lesssim n^\alpha$ for some $0 < \alpha < 1$. This is slower than the near-optimal concentration rate $n^{-\frac{1}{2}} \sqrt{K_{f_0} \log^{2\eta}(n/K_{f_0})}$, if we use the softplus link function, instead. This demonstrates the need for choosing suitable link functions in empirical applications.

# 6 DISCUSSION

In this paper we have examined a general framework for Bayesian Additive Regression Tree Models that encapsulates various conventional BART models adapted to a wide range of regression and classification tasks. We demonstrated that these models have a near-minimax posterior concentration rate for a wide range of functions, thus corroborating the empirical success of BART and its variants, from a theoretical perspective. These results also build the foundation for uncertainty quantification statements for a wide variety of BART models, opening up interesting avenue for future research. The theoretical results also substantiate the scope of a wider variety of distributions on approximating step-heights, that can prove advantageous for applications where the response distribution has a thicker tail. These theoretical findings also provide strong motivation for exploring novel application areas for flexible BART-like models. It is worth noting that our results are based on a modified version of the original BART prior Chipman et al. (1998). The reason behind this modification is that the original prior does not decay at a fast enough rate. However, since we examine only sufficient (but not necessary) conditions for optimal posterior concentration, our results do not guarantee that the original prior is inherently worse than the modified prior. The original BART prior will be examined in future work. We did not provide any empirical examples as these have been studied using existing BART software packages (e.g. BART in R and PyMC-BART in Python).

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

# A   APPENDIX

## A.1   BAYESIAN CART PRIOR BY DENISON ET AL. (1998)

We describe the Bayesian CART prior proposed by Denison et al. (1998). The prior on individual Bayesian trees is assigned conditional on the number of terminal nodes/ leaves $K$ and all prior probability is concentrated on the set of all *valid* tree partitions, as defined below (Definition 3.1 of Rockova et al. (2020)):

**Definition A.1.** Denote by $\mathbf{\Omega} = \{\Omega\}_{k=1}^{K}$, a partition of $[0,1]^p$, We say that $\mathbf{\Omega}$ is valid if

$$\mu(\Omega_k) \geq \frac{C}{n} \quad \forall k = 1, \ldots, K \tag{12}$$

for some $C \in \mathbb{N} \setminus \{0\}$.

Valid partitions have non-empty cells, where the allocation does not need to be balanced. Now the prior on tree partitions is specified as follows:

1. The number of leaves in a tree $K$ follows a Poisson distribution with parameter $\lambda > 0$

$$P(K) = \frac{\lambda^K}{(e^\lambda - 1)K!}, \qquad k = 1, 2, \ldots \tag{13}$$

2. Given the number of leaves $K$, a tree is chosen uniformly at random from the set of all available *valid* tree-partitions with $K$ leaves. Number of valid tree partitions is given by

$$\Delta(V_K) = \frac{q^{K-1}n!}{(n-K+1)!} \tag{14}$$

This is a slightly modified version of the original prior proposed by Denison et al. (1998). This modified version was used by Rockova et al. (2020) to derive posterior concentration rates for the BART estimator under this prior.

3. At each node, the splitting rule consists of picking a split variable $j$ uniformly at random from the available directions $\{1, \ldots, q\}$ and picking a split point $c$, also uniformly at random from the available data values $x_{1j}, \ldots, x_{nj}$.

## A.2   PRELIMINARY RESULTS WITH PROOF

**Lemma A.2.** *The multivariate Gaussian $\mathcal{N}_p(\mathbf{0}, \mathbb{I}_p)$ and the multivariate Laplace $\mathcal{L}_p(\mathbf{0}, \mathbb{I}_p)$ distribution belong to the general family of distributions with CDF $F_\beta$ that has the following property: For some $C_1 > 0$, $0 < C_2 \leq 2$ and $C_3 > 0$ and any $t > 0$,*

$$F_\beta(\|\beta\|_\infty \leq t) \gtrsim \left(e^{-C_1 t^{C_2}}t\right)^p \quad for\ t > 0 \tag{15}$$

$$F_\beta(\|\beta\|_\infty \geq t) \lesssim e^{-C_3 t} \quad for\ t \geq 1 \tag{16}$$

*Proof.* If $F_\beta = \mathcal{N}_p(\mathbf{0}, \mathbb{I}_p)$, then for any $t > 0$,

$$F_\beta(\|\beta\|_\infty \leq t) \gtrsim \left(e^{-t^2/2}\int_{-t}^{t} d\beta\right)^p \gtrsim e^{-pt^2/2}t^p$$

For $t \geq 1$

$$F_\beta(\|\beta\|_\infty \geq t) \lesssim \left(e^{-t^2/4}2\int_{t}^{\infty} e^{-z^2/4}dz\right)^p \lesssim e^{-C_3 t}$$

If $F_\beta = \mathcal{L}_p(\mathbf{0}, \mathbb{I}_p)$, then for any $t > 0$,

$$F_\beta(\|\beta\|_\infty \leq t) \gtrsim \left(e^{-t}\int_{-t}^{t} d\beta\right)^p \gtrsim e^{-pt}t^p$$

Also, for any $t \geq 0$,

$$F_\beta(\|\beta\|_\infty \geq t) = \frac{e^{-pt}}{2} < e^{-pt}$$

$\square$

**Lemma A.3.** *Let $f$ and $f_0$ denote step functions of the form $f(\boldsymbol{X}) = \sum_{k=1}^{K} \beta_k \mathbb{I}(\boldsymbol{X} \in \Omega_k)$ and $f_0(\boldsymbol{X}) = \sum_{k=1}^{K} \beta_k^0 \mathbb{I}(\boldsymbol{X} \in \Omega_k)$ respectively, on a tree-shaped partition $\{\Omega_k\}_{k=1}^{K}$. Let $P_f$ and $P_{f_0}$ denote two probability densities belonging to an Exponential family distribution of the form*

$$P_f(\boldsymbol{Y} \mid \boldsymbol{X}) = h(\boldsymbol{Y})g\left[f(\boldsymbol{X})\right]\exp\left[\eta\left(f(\boldsymbol{X})\right)^T T(\boldsymbol{Y})\right], \tag{17}$$

*with parameters $f$ and $f_0$ respectively. If $\left|\frac{\nabla^T g(\boldsymbol{\beta})}{g(\boldsymbol{\beta})}\right| \leq C_g^n \boldsymbol{1}_p$, for some positive sequence $\{C_g^n\}_{n\geq 1}$, then*

$$K_n(P_f, P_{f_0}) \vee V_n(P_f, P_{f_0}) \lesssim C_g^n \sum_{k=1}^{K} \left\|\beta_k - \beta_k^0\right\|_1 \tag{18}$$

$$H_n(P_f, P_{f_0}) \lesssim C_g^n \sum_{k=1}^{K} \left\|\beta_k - \beta_k^0\right\|_1 \tag{19}$$

*Proof.* Denoting $f_i = f(\boldsymbol{X}_i)$ and $f_{i0} = f_0(\boldsymbol{X}_i)$, we can write

$$
\begin{aligned}
&K_n(P_f, P_{f_0}) \\
&= \frac{1}{n} \sum_{i=1}^{n} g(f_i) \int h(\boldsymbol{Y}) \exp\left(f_i T(\boldsymbol{Y})\right) [\log \frac{g(f_i)}{g(f_{i0})} \\
&\quad + \exp\left[(f_i - f_{i0})^T T(\boldsymbol{Y})\right]] d\boldsymbol{Y} \\
&= \frac{1}{n} \sum_{i=1}^{n} \left[\log \frac{g(f_i)}{g(f_{i0})} + (f_i - f_{i0})^T \mathbb{E}\left[T(\boldsymbol{Y})\right]\right] \\
&= \sum_{k=1}^{K} \mu(\Omega_k) \left[\log \frac{g(\beta_k)}{g(\beta_k^0)} - \frac{\nabla^T g(\beta_k)}{g(\beta_k)}\left(\beta_k - \beta_k^0\right)\right]
\end{aligned}
$$

By triangle inequality and Taylor series approximation of $\log \nabla g(\beta_k)$ about $\beta_k^0$, we get

$$K_n(P_f, P_{f_0}) \lesssim \sup \left|\frac{\nabla^T g(\cdot)}{g(\cdot)}\right| \sum_{k=1}^{K} \left\|\beta_k - \beta_k^0\right\|_1$$

$$= C_g^n \sum_{k=1}^{K} \left\|\beta_k - \beta_k^0\right\|_1,$$

Similar technique works for $V_n(P_f, P_{f_0})$

Also, Since Hellinger metric is bounded from above by Kullback-Leibler divergence, $H_n$ satisfies,

$$H_n(P_f, P_{f_0}) \lesssim C_g^n \sum_{k=1}^{K} \left\|\beta_k - \beta_k^0\right\|_1$$

$\square$

**Lemma A.4.** *Any bounded **monotone** function $f_0$ can be approximated with arbitrary precision $\varepsilon_n$, by a step function supported on a k-d tree partition with $\widehat{K} \geq \lceil 1/\epsilon_n \rceil$ leaves.*

*Proof.* Without loss of generality, assume $0 \leq f_0(\cdot) \leq 1$. Partition interval $[0, 1]$ by $0 = y_0 < y_1 < \cdots < y_k < \cdots < y_{K-1} < y_K = 1$, with $K = \lceil 1/\epsilon_n \rceil$. Then $\left| y_k - y_{k-1} \right| < \varepsilon_n$ and we can approximate $f_0(\boldsymbol{X})$ by the step function:

$$f(\boldsymbol{X}) = \sum_{k=1}^{K} y_k \mathbb{I}\{\boldsymbol{X} \in \Omega_k\}$$

, where $\Omega_k = f^{-1}[y_{k-1}, y_k]$.

If $f$ is monotone, $\Omega_k = \prod_{j=1}^q \{x_j \in I_j\}$, where $I_j$ is an interval and $x_j$ denotes the $j$-th coordinate of $\boldsymbol{X} \in \mathbb{R}^q$.

Since any step function supported on an axis-paralleled partition has an equivalent step function supported on a k-d tree, we can approximate the axis paralleled partition $\{\Omega_k\}_{k=1}^K$ by a recursive binary tree partition $\{\widehat{\Omega}_k\}_{k=1}^{\widehat{K}}$ with number of leaves $\widehat{K} \geq K$. $\qquad\square$

### A.3  PROOF OF MAIN RESULTS

In this section we prove Theorem 4.1 and Theorem 4.3. Most steps in the proofs are identical and hence for simplicity we describe the common steps of the proofs together and mark the steps that are different by the corresponding theorem number. We need to prove three conditions: entropy condition (C1), prior concentration condition (C2) and prior decay rate condition (C3). The steps of the proofs for each of these conditions are described below.

#### A.3.1  ENTROPY CONDITION

Define
$$\mathcal{F}_n = \{f_{\boldsymbol{T},\boldsymbol{\beta}}(\boldsymbol{X}) \text{ of the form equation 2 with } K = k_n \text{ and } \|\beta\|_\infty \leq C_\beta^n\},$$
where $k_n \propto n\varepsilon_n^2 / \log n$ and $C_\beta^n$ is defined in Assumption 1.

Since $\|\boldsymbol{z}\|_1 \leq Kp\|\boldsymbol{z}\|_\infty$ for any $\boldsymbol{z} \in \mathbb{R}^{Kp}$, by the bound equation 19 and by definition of $\mathcal{F}_n$, we can write
$$N\left(\frac{\varepsilon_n}{36}, \mathcal{F}_n, H_n\right) \lesssim \sum_{K=1}^{k_n} N\left(\frac{\varepsilon_n}{36 C_g^n Kp}, \{\beta : \|\beta\|_\infty \leq C_\beta^n\}, \|\cdot\|_\infty\right)$$
$$\lesssim \sum_{K=1}^{k_n} \left(\frac{36 C_\beta^n C_g^n Kq}{\varepsilon_n}\right)^{Kq}$$

Therefore the LHS of (C1) can be bounded from above by
$$(k_n + 1)p \left[\log 36 + \log(C_\beta^n C_g^n) + \log k_n + \log p - \log \varepsilon_n\right]$$

Since $C_\beta^n C_g^n \lesssim n^M$ for some $M > 0$, ignoring smaller terms, proving condition (C1) reduces to proving
$$(k_n + 1)p \log n \lesssim n\varepsilon_n^2 \tag{20}$$

**Theorem 4.1:**  When $f_0$ is a step function with complexity $K_{f_0}$ we can prove equation 20 by replacing $\varepsilon_n = n^{-1/2}\sqrt{K_{f_0} \log^{2\eta}(n/K_{f_0})}$ and $k_n \propto \frac{n\varepsilon_n^2}{p\log(n/K_{f_0})} = K_{f_0} \log^{2\theta-1}(n/K_{f_0})$ for some $\theta > 1/2$.

**Theorem 4.3:**  When $f_{0l}$ is a $\nu$-Hölder continuous function with $0 < \nu \leq 1$ for all $l = 1, \ldots, p$, replacing $\varepsilon_n = n^{-\nu/(2\nu+q)}\sqrt{\log n}$ and $k_n \propto \frac{n\varepsilon_n^2}{\log n} = n^{q/(2\nu+q)}$ proves equation 20.

#### A.3.2  PRIOR CONCENTRATION CONDITION

Let $\widetilde{f_0} = \left(f_{\boldsymbol{T},\boldsymbol{B}_1^0}(\boldsymbol{x}), \ldots, f_{\boldsymbol{T},\boldsymbol{B}_{q-1}^0}(\boldsymbol{x})\right)$ denote the projection of $f_0$ onto a balanced k-d tree partition with $a_n$ leaves, where $a_n$ is chosen so that $\left\|f_0 - \widetilde{f_0}\right\|_{2,n} < \varepsilon_n/2$.

**Theorem 4.1:**  If $f_0$ is a step function, $a_n = K_{f_0}$

**Theorem 4.3:**  If $f_0$ is a $\nu$-Hölder continuous function, $a_n$ is chosen by the following lemma, which is analogous to Lemma 3.2 of Rockova et al. (2020).

**Lemma A.5.** *Denote $f = \{f_l\}_{l=1}^p$ and assume $f_l \in \mathcal{H}^{\nu_l}$ where $\nu_l \leq 1$ for all $l = 1, \ldots, p$ and $\mathcal{X}$ is regular. Then there exists tree structured step functions $\hat{f} = \{f_{\boldsymbol{T}, \boldsymbol{B}_l}\}_{l=1}^p \in \mathcal{F}_K$ for some given tree partition $\boldsymbol{T}$ with $K \in \mathbb{N}$ leaves such that for some constant $C > 0$,*

$$\left\|\hat{f} - f\right\|_{2,n} \leq Cd \sum_{l=1}^p \left(\frac{1}{K^{\nu_l/q}} \|f_l\|_{\mathcal{H}^{\nu_l}}\right) \leq C \frac{q}{K^{\nu/q}} \sum_{l=1}^p \left(\|f_l\|_{\mathcal{H}^{\nu_l}}\right),$$

*where $\nu = \min_{l=1}^p \nu_l$.*

As a corollary, replacing $C_0 = C\left(\sum_{l=1}^p \|f_l\|_{\mathcal{H}^\nu}\right)$, $a_n$ satisfies

$$\left(\frac{2C_0 q}{\varepsilon_n}\right)^{q/\nu} \leq a_n \leq \left(\frac{2C_0 q}{\varepsilon_n}\right)^{q/\nu} + 1 \tag{21}$$

Using equation 18 and by triangle inequality, we can bound the LHS of (C2) from below by

$$C\pi(a_n)\Pi\left(\beta \in B_n^{a_n} : \left\|\beta - \beta^0\right\|_1 \leq \frac{\epsilon_n^2}{2C_g^n}\right)$$

For the prior by Chipman et al. (2010), $C = 1$ and $\pi(a_n) \gtrsim e^{-a_n \log a_n}$ (by Corollary 5.2 of Rockova & Saha (2019)).

For the prior by Denison et al. (1998), $C = \frac{1}{|F_{a_n}|} > (a_n dn)^{-a_n} > e^{-a_n \log a_n}$ (by Lemma 3.1 of Rockova et al. (2020)) and $\pi(a_n) \gtrsim e^{-a_n \log a_n}$ (by proof of Theorem 4.1 of Rockova et al. (2020)).

Thus for both priors $C\pi(a_n) \gtrsim e^{-2a_n \log a_n}$.

Next we bound $\Pi\left(\beta \in B_n^{a_n} : \left\|\boldsymbol{\beta} - \boldsymbol{\beta}^0\right\|_1 \leq \frac{\epsilon_n^2}{2C_g^n}\right)$, up to a constant, from below by

$$\Pi\left(\boldsymbol{\beta} : \|\boldsymbol{\beta}\|_\infty \leq C_\beta^n, \quad \left\|\boldsymbol{\beta} - \boldsymbol{\beta}^0\right\|_\infty \leq \frac{\epsilon_n^2}{2a_n q C_g^n}\right)$$

Since $C_g^n$ and $C_\beta^n$ both are increasing with $n$, for sufficiently large $n$, the above expression is bounded below by

$$\Pi\left(\boldsymbol{\beta} : \left\|\boldsymbol{\beta} - \boldsymbol{\beta}^0\right\|_\infty \leq \frac{\varepsilon_n^2}{2a_n p C_g^n}\right)$$

$$\gtrsim e^{-C_1 a_n p\left(\|\beta_0\|_\infty + \frac{\varepsilon_n^2}{2a_n p C_g^n}\right)_2^C} \left(\|\beta_0\|_\infty + \frac{\varepsilon_n^2}{2a_n p C_g^n}\right)^{a_n p}$$

Since $\varepsilon_n^2 \to 0$ and both $a_n$ and $C_g^n$ are both increasing with $n$, assuming $\|f_0\|_\infty \lesssim \sqrt{\log n}$, the above bound reduces to

$$e^{-C_1 a_n p \log^{C_2/2} n} \|\beta_0\|_\infty^{a_n p/2} \gtrsim \log\left[-C_1 a_n p \log^{C_2/2} n\right]$$

We can prove $e^{-a_n \log n} \gtrsim e^{-n\varepsilon_n^2}$ for Theorem 4.1 and Theorem 4.3 separately by replacing appropriate values of $\varepsilon_n$. Since $C_2 \leq 2$, this would complete the proof.

### A.3.3 PRIOR DECAY RATE CONDITION

**Theorem 4.1:** When $f_0$ is a step-function with complexity $K_{f_0}$,

$$\Pi(\mathcal{F} \setminus \mathcal{F}_n) \leq \Pi(\mathcal{F} \setminus \bigcup_{K=1}^{k_n} F_K) + \Pi(\bigcup_{K \leq k_n} \{f \in F_K : \|\beta\|_\infty > C_\beta^n\})$$

$$\leq \Pi(\bigcup_{K > k_n} F_K) + e^{-K_{f_0} \log n/2}$$

$$= \Pi(\bigcup_{K > k_n} F_K) + o(e^{-n\varepsilon_n^2})$$

The last line is due to the fact $C_\beta^n \gtrsim K_{f_0} \log n$ when $f_0$ is a step-function with complexity $K_{f_0}$.

**Theorem 4.3:** When $f_0$ is a $\nu$-Hölder continuous function, the LHS of condition (C3) can be bounded from above by

$$\Pi(\mathcal{F} \setminus \mathcal{F}_n) \leq \Pi(\mathcal{F} \setminus \bigcup_{K=1}^{k_n} F_K) + \Pi(\bigcup_{K \leq k_n} \{f \in F_K : \|\beta\|_\infty > C_\beta^n\})$$

$$\leq \Pi(\bigcup_{K > k_n} F_K) + \sum_{K=1}^{k_n} \Pi(\{\beta : \|\beta\|_\infty > C_\beta^n\})$$

$$\leq \Pi(\bigcup_{K > k_n} F_K) + \sum_{K=1}^{k_n} e^{-C_\beta^n}, \quad \text{by condition equation 7}$$

$$\leq \Pi(\bigcup_{K > k_n} F_K) + k_n e^{-C_\beta^n}$$

$$= \Pi(\bigcup_{K > k_n} F_K) + o(e^{-n\varepsilon_n^2})$$

The last line is due to the fact $C_\beta^n \gtrsim n$, when $f_0$ is a $\nu$-Hölder continuous functions.

Therefore it is enough to show that

$$\Pi(\bigcup_{K > k_n} F_K) \lesssim e^{-n\varepsilon_n^2}$$

This condition is satisfied for both priors under consideration. This follows from section 8.3 of Rockova et al. (2020) for the prior by Denison et al. (1998) and from Corollary 5.2 of Rockova & Saha (2019) for the prior by Chipman et al. (2010).

### A.4 CLASSIFICATION WITH DIRICHLET STEP HEIGHTS

For a multi-class classification problem with $p$ classes, where the response variable $\boldsymbol{Y}$ is a categorical random variable with $p$ categories, $\boldsymbol{Y}$ can be written as a $p$ dimensional binary vector that has $1$ at the $l$-th coordinate if $\boldsymbol{Y}$ belongs to category $l \in \{1, \ldots, p\}$ and $0$ elsewhere. G-BART assumes

$$\boldsymbol{Y} \mid \boldsymbol{X} \sim \text{Multinomial}(1, \boldsymbol{f_0}(\boldsymbol{X})), \tag{22}$$

where $\boldsymbol{f_0} = (f_{01}, \ldots, f_{0p})' : \mathbb{R}^q \to (0, 1)^p$ is a constrained function with $\boldsymbol{f_0}(\boldsymbol{X})' \boldsymbol{1}_p = 1$ for any $\boldsymbol{X} \in \mathbb{R}^q$. Each $f_{0l}(\cdot)$ can be approximated by a step function of the form

$$f_{\boldsymbol{T}, P}(\boldsymbol{x}) = \sum_{k=1}^{K} P_k \mathbb{I}(\boldsymbol{x} \in \Omega_k) \tag{23}$$

on a tree-partition $\{\Omega_k\}_{k=1}^{K}$. Denison et al. (1998) assumes

$$P_k = (P_{k1}, \ldots, P_{kp}) \stackrel{i.i.d}{\sim} \text{Dirichlet}(\alpha_1, \ldots, \alpha_p), \tag{24}$$

where $\alpha_l > 0, \quad \forall l \in \{1, \ldots, p\}$.

**Theorem A.6.** *If we assume that the distribution of the step-sizes satisfies equation 24, then under Assumptions 1 & 2 described in section 4 of the manuscript, the Bayesian Tree estimator satisfies the following property,:*

*(i) If $f_0$ is $\nu$-Hölder continuous with $0 < \nu \leq 1$ where $\|f_0\|_\infty \lesssim \log^{1/2} n$, then with $\varepsilon_n = n^{-\alpha/(2\alpha+p)} \log^{1/2} n$, and*

*(ii) If $f_0$ is step-function with complexity $K_{f_0} \lesssim \sqrt{n}$, then with $\varepsilon_n = n^{-1/2}\sqrt{K_{f_0} p \log^{2\nu}(n/K_{f_0}p)n}$,*

$$\Pi\left(f \in \mathcal{F} : H_n(\mathbb{P}_f, \mathbb{P}_{f_0}) > M_n \varepsilon_n \mid \boldsymbol{Y}^{(n)}\right) \to 0,$$

*for any $M_n \to \infty$ in $\mathbb{P}_{f_0}^{(n)}$-probability, as $n, p \to \infty$.*

*The above statement is true for both tree priors considered in this paper: the prior by Denison et al. (1998) and a modified version of the prior by Chipman et al. (1998) with $p_{split}(\Omega_t) = \alpha^{d(\Omega_t)}$ for some $1/n \leq \alpha < 1/2$.*

*Proof.* We need to prove three conditions: entropy condition (C1), prior concentration condition (C2) and prior decay rate condition (C3). Among these (C1) and (C3) can be proved by the same technique as in section A.3. Therefore we will only prove Condition (C3) here. We need to show, for some $c > 0$

$$\Pi\left(f \in \mathcal{F} : max\{K_n(f, f_0), V_n(f, f_0)\} \leq \varepsilon_n^2\right) \gtrsim e^{-cn\varepsilon_n^2} \tag{25}$$

Let $\widetilde{f}_0 = \left(f_{\boldsymbol{T}, P_1^0}(\boldsymbol{x}), \ldots, f_{\boldsymbol{T}, P_q^0}(\boldsymbol{x})\right)$ denote the projection of $f_0$ onto a balanced k-d tree partition $\boldsymbol{T}$ with $a_n$ leaves, where $a_n$ is chosen so that $\left\|f_0 - \widetilde{f}_0\right\|_{2,n} < \varepsilon_n/2$. If $f_0$ is a step function, $a_n = K_{f_0}$. If $f_0$ is a $\nu$-Hölder continuous function, $a_n$ is chosen by Lemma 3.2 of Rockova et al. (2020), where replacing $C_0 = C\left(\sum_{l=1}^{p} \|f_l\|_{\mathcal{H}^\nu}\right)$ we get

$$\left(\frac{2C_0 q}{\varepsilon_n}\right)^{q/\nu} \leq a_n \leq \left(\frac{2C_0 q}{\varepsilon_n}\right)^{q/\nu} + 1 \tag{26}$$

$f_{\boldsymbol{T}, P_l^0}(\boldsymbol{x})$ is of the form equation 23 for some tree topology $\boldsymbol{T}$ with $a_n$ leaves and $P_l^0 = \{P_{kl}^0\}_{k=1}^{a_n}$ for $l = 1, \ldots, p$. We assume there exists some $\delta_0 > 0$ such that $\min f_{0l} > \delta_0$ for all $l = 1, \ldots, q$. This implies $P_{lk}^0 > \delta_0$ for all $l = 1, \ldots q$ and all $k = 1, \ldots, K$. Therefore by equation 18, we can bound the LHS of equation 25 from above by

$$C\pi(a_n)\Pi\left(P \in [0,1]^{a_n p} : \left\|P - P^0\right\|_1 \leq \delta_0 \varepsilon_n^2/2\right)$$

For the prior by Chipman et al. (1998), $C = 1$ and for the prior by Denison et al. (1998), $C = \frac{1}{|F_{a_n}|} > (a_n d n)^{-a_n} > e^{-a_n \log a_n}$ (by Lemma 3.1 of Rockova et al. (2020)). By Corollary 5.2 of Rockova & Saha (2019) for the prior by Chipman et al. (1998) and by proof of Theorem 4.1 of Rockova et al. (2020) for the prior by Denison et al. (1998), we can show $\pi(a_n) \geq e^{-a_n \log a_n}$. Thus for both priors,

$$C\pi(a_n) > e^{-2a_n \log a_n} \tag{27}$$

Since $P_k \sim \text{Dirichlet}(\alpha_1, \ldots, \alpha_p)$ for all $k = 1, \ldots, K$ and $P_{lk}^0 > \delta_0$, for all $l = 1, \ldots, p$ and all $k = 1, \ldots, K$, we can bound $\Pi\left(P \in [0,1]^{a_n q} : \left\|P - P^0\right\|_1 \leq \delta_0 \epsilon_n^2/2\right)$ from above by

$$\Pi\left(P \in [0,1]^{a_n p} : \left\|P - P^0\right\|_\infty \leq \frac{\delta_0 \epsilon_n^2}{2a_n p}\right) \gtrsim C_\alpha(\frac{\delta_0 \epsilon_n^2}{a_n p})^{a_n p}, \tag{28}$$

where $C_\alpha$ is a constant that depends on the Dirichlet parameters $\alpha = (\alpha_1, \ldots, \alpha_q)$. Combining equation 27 and equation 28 completes the proof. □

PROOF OF THEOREM 4.3

The first step is to find an approximating step-function $\widehat{f}_0$ by Lemma 4.2, such that $\|f_0 - \widehat{f}_0\|_{2,n} < \varepsilon_n/2$. The rest of the proof follows by retracing the steps as in the proof of Theorem 4.4 given above.

A.5   PARSIMONY OF G-BART

The following statements support the empirical observation that generalized Bayesian trees are resilient to overfitting. The proofs of (i), (ii) and (iii) follow from Lemma 1 of Ghosal et al. (2007), in conjunction with the proofs of Theorems 4.1, 4.3 and 4.4 respectively.

**(i)** Under the assumptions of Theorem 4.1 we have $\Pi\left(K \gtrsim K_{f_0} \mid \boldsymbol{Y}^{(n)}\right) \to 0$ in $\mathbb{P}_{f_0}^{(n)}$-probability, as $n, q \to \infty$.

**(ii)** Under the assumptions of Theorem 4.3 we have $\Pi\left(K \gtrsim n^{q/(2+q)} \mid \boldsymbol{Y}^{(n)}\right) \to 0$ in $\mathbb{P}_{f_0}^{(n)}$-probability, as $n, q \to \infty$.

**(iii)** Under the assumptions of Theorem 4.4 we have $\Pi\left(K \gtrsim n^{q/(2\nu+q)} \mid \boldsymbol{Y}^{(n)}\right) \to 0$ in $\mathbb{P}_{f_0}^{(n)}$-probability, as $n, q \to \infty$.

