# OpenReview forum: "Bayesian Additive Regression Trees for Exponential Family Distributions: A Theoretical Perspective"
_ICLR.cc/2026/Conference — Submitted to ICLR 2026_

### Official Review · Reviewer_5DxC · 2025-10-18

**Soundness:** 3
**Presentation:** 2
**Contribution:** 2
**Rating:** 4
**Confidence:** 2

**Summary:**

BART is a Bayesian ensemble tree model that is widely used as a method for flexibly estimating nonlinear functions while quantifying prediction uncertainty. While previous theoretical research has been limited to Gaussian responses (continuous variables), this paper formulates Generalized BART (G-BART), which handles more general response variables that follow an exponential family, and theoretically analyzes the posterior concentration rate of its posterior distribution.

**Strengths:**

Originality: This is the first attempt to generalize BART to general exponential families.
Quality: It is a theoretical paper. Each mathematical steps seem correct.
Clarity: What is done is clear.
Significance: The posterior concentration rate  indicates how quickly the posterior distribution converges around the true function $f_0$ as the number of samples increases. Based on theorem by Ghosal et al. (2007), the authors rigorously evaluate the posterior concentration rate for three classes of $f_0$; i) step functions, ii) monotone functions, and iii) Hoelder continuous functions.
The results guarantee that the posterior converges approximately consistently to the minimax rate for the true function, providing a theoretical justification for BART's empirical success. The results would provides related fields with a certain impact.

**Weaknesses:**

The definitions of "natural parameters" and "link functions" are incorrect. In this paper, the "expectation parameters" are called "natural parameters", and the "inverse function of the link function" is called "link function", which was confusing. The idea of ​​extending BART, which was defined for normal distributions, to generalized linear models is incremental and not revolutionary. It is purely theoretical analysis, so it is unclear how useful it is for real-world data analysis.

**Questions:**

The results obtained in this paper require the following conditions: i) Entropy condition (controlling the complexity of the model space), ii) Sufficient prior probability (sufficient mass in the true neighborhood), iii) Prior damping condition (low probability in unnecessary regions).
How reasonable is it in practical situations to assume these conditions?

---

### Official Review · Reviewer_WH5f · 2025-10-30

**Soundness:** 3
**Presentation:** 3
**Contribution:** 2
**Rating:** 4
**Confidence:** 5

**Summary:**

This paper demonstrates the posterior contraction of the BART model using classical theoretical pipelines in nonparametric Bayesian analysis. The key aspects of the proof and validation are the priors on the depth and partition of the trees, which are designed elaborately. The mathematical tools and experiments validate the success of proofs, and it seems successful. However, the implication of validating theoretical results is not clear in the advancement of core algorithms or applications in the fields of BART.

**Strengths:**

The paper considers various models,, including thethe  GLM-type link functios and, monotone functions, among othersn. The contraction rates achieve the mini-max rate, and the variation by the modes is well-calibrated. Classical techniques for the posterior contraction rates are well-suited for proofs.

**Weaknesses:**

Although the theoretical validation is interesting, the implications of these findings are not immediately apparent. It only provides the validation in the use of BART. As I know, the use of BART is widespread, and the properties of BART cannot provide new evidence in an empirical domain for a new algorithm or others.

**Questions:**

1. In theorem 4.4, the effect of q is not negligible in some cases, do you have any ides to reduce the convergence rate with a situation that the covariates are correlated.
2. Can you have comparison results, such as the difference between the original BART prior Chipman et al. (1998) and the proposed one?

**Details Of Ethics Concerns:**

None.

---

### Official Review · Reviewer_GTnX · 2025-11-01

**Soundness:** 3
**Presentation:** 2
**Contribution:** 2
**Rating:** 2
**Confidence:** 3

**Summary:**

The authors derive posterior concentration results for the BART model under two main modifications:
1. The true distribution of the response is **multivariate exponential**.
2. The distribution at each leaf can belong to a **broader family** than the Gaussian distribution commonly used in BART.

The concentration results are derived for three forms of the exponential distribution of the response, depending on the functional form of \( f_0 \).

**Strengths:**

- Novel posterior concentration results for **multivariate BART**.
- The authors study **multiple generative processes** for the response.

**Weaknesses:**

- The authors do **not discuss the BART sampler** for any of the new variants they establish rates for.
- [1] already establishes **concentration results for univariate generalized BART**, limiting the novelty of this analysis.
- **No empirical study or implementation** of the analyzed algorithm (important since the results are asymptotic).

**Questions:**

1. Can the authors clearly position their work against [1]?
2. Has the algorithm currently studied **ever been used or implemented**?
3. Can the authors provide a **description of their proposed sampler**? How is it different from the practical BART sampler?
4. How does the **concentration depend on the dimension** of \( y \) and \( p \)?


[1] Linero, A.R., 2025. Generalized Bayesian additive regression trees models: Beyond conditional conjugacy. Journal of the American Statistical Association, 120(549), pp.356-369.

---

### Official Review · Reviewer_xCs6 · 2025-11-08

**Soundness:** 3
**Presentation:** 2
**Contribution:** 3
**Rating:** 6
**Confidence:** 3

**Summary:**

The paper proposes a generalized BART (G‑BART) theory for exponential‑family responses and proves near‑minimax posterior concentration in three regimes: step functions, monotone functions, and Hölder($0<\nu\le1$) functions. The proofs follow the Ghosal–Ghosh–vdV sieve template with sufficient conditions that are portable across likelihoods via constraints on the exponential‑family factor g and step‑height priors. A notable practical outcome is link‑function guidance for Poisson regression (softplus vs exponential).

**Strengths:**

The authors explicitly show softplus yields near‑minimax rates while the exp link can slow the rate for Poisson. This is positioned as an implication of their framework, not as a generic BART fact. This result could be elevated from good to great by testing the fit of BART with both link functions and seeing whether there are any meaningful differences in the performance on the algorithm. Even if on a toy example, this would give practitioners valuable guidance and should expand the scope and impact of the paper.

The assumptions are reasonable as the authors show that they hold for distributions like Gaussian and Laplace, and are transparent about the fact that the conditions are sufficient but not necessary (and even give a counterexample).

Many of the standard known results for BART fall out as corollaries to this paper, which is nice.

**Weaknesses:**

The paper does a respectable job flagging its own contributions and acknowledging when assumptions are merely sufficient, but it sometimes blurs the line between what’s standard (vdV/Ghosal sieve machinery, known tree‑prior mass/tail bounds) and what’s new (the exponential‑family checklist, link‑function guidance). A few inconsistencies in the prior specification and some “placeholder” references make the novelty/boilerplate boundary less crisp than it could be.

Please add a Section 1.1 OUR CONTRIBUTIONS or explicitly list what’s new. In my view it is

(i) moving beyond Gaussian to an exponential‑family response with sufficient conditions for posterior concentration,
(ii) replacing Gaussian step heights by a family (e.g., Laplace) under small‑ball/tail conditions, and
(iii) covering non‑Hölder targets (step/monotone).
(iv) Question: Mathematically are there any other innovations/proof techniques that are not standard in the literature?
(v) Question: What else? Did I miss something?

There are typos like see section ?? and

“Sof tmax” $\to$“Softmax”

“Althogh” $\to$  “Although”

You first cite prior work using the Hölder smoothness letter $\alpha$ then use letter $\nu$. Please stick with one convention throughout, I suggest using $\nu$ since $\alpha$ is already used in the prior. Aside from these expositional and formatting issues, I couldn't find any major mistakes in the paper.

**Questions:**

The paper presents a standard posterior contraction argument that uses three vdV/Ghosal conditions (C1)–(C3) on an increasing sequence of sieves. Then it seems like under (a) identity link $\eta(z) = z$, we require a box $B_{n} :=  \mathcal{B}(0, C_\beta^{n}, \ell^{\infty})$ under which $|| \nabla \log{g}(\beta) || \leq C_{g}^{n} \cdot 1_{p}$. This boils down to assuming that $\log{g}$ is Lipschitz on a growing box. In particular, all dependence on the particular likelihood will flow through the log of the partition function. Because from the partition function we can get all moments of the sufficient statistic, is there any way under which the conditions of the theorem could be stated in terms of the moments of the exponential family?

The paper uses

$$p_{split}(\Omega) \propto \alpha^{-d(\Omega)}, \quad \alpha \in [0,1/2)$$

this increases with depth while the supplement uses

$$p_{split}(\Omega_{t}) = \alpha^{d(\Omega_{t})}$$

decreasing with depth. It seems like the proofs need decreasing splits. This could be confusing, please fix Eq. (5) to depth-decreasing and remove the "see section ??"

What exactly is “modified” in the Denison prior is not isolated in one place. Appendix A.1 calls it a “slightly modified” version used by Rocková et al. (2020), and gives the counting of valid partitions $\Delta(\mathcal V_K),$ but the text doesn’t succinctly enumerate which pieces differ from Denison (1998) and why those modifications are needed here. Readers must infer this from citations, which muddies the standard/new line.

Appendix A.3 opens with “Most steps in the proofs are identical… we describe the common steps” and then carries out the (C1)–(C3) checks. It uses standard calculus/covering tricks (e.g., the $\beta$-box covering leading to $(k_n+1)^p\log n\lesssim n\varepsilon_n^2$), but doesn’t explicitly label the key novel technical steps. Are these the exponential‑family linearization that collapse KL/Hellinger/Var to an \ell_1 bound with a C_g^n factor? As written, a non‑expert could miss which inequalities are “just vdV/Ghosal” vs “new to this paper.”

I think it will improve the flow, especially for non-experts, to remind the reader about the standard "template" for posterior concentration arguments and briefly discuss the intuition behind them (I found myself having to review vdV and Ghosal).

I am leaning towards weak accept, but I am willing to raise my score to accept if convincing experiments on Poisson BART are consistent with the theory. (i.e., comparing softplus vs exp links).

---

### Meta-Review · Area_Chair_pvf5 · 2025-12-22

**Summary:**

This paper addresses a generalized framework for Bayesian additive regression trees where the response variable comes from an exponential family distribution.  They derive sufficient conditions on the response distribution, under which the posterior concentrates at a near minimax rate. This paper aims to bridge the gap between empirical success for non-Gaussian and theoretical justification mainly for Gaussian, by proving that BART maintains near-optimal performance theoretically even when applied to general exponential family distributions. There are critical concerns raised by reviewers. First of all, it should be clarified what's new in this paper and what are available in existing work. A recent work [1] on concentration results for univariate generalized BART is missing, which limits the novelty of this paper. There is no empirical study to demonstrate the behavior or performance of the proposed method. Therefore, the paper is not recommended for acceptance in its current form. I hope authors found the review comments informative and can improve their paper by addressing these carefully in future submissions.

**Reviewer Concerns:**

The critical concerns summarized above remain unaddressed in the authors' response.

**Reviewer Scores:**

All reviewers are expected to maintain their original scores, as no author response was submitted.

---

### Decision · Program_Chairs · 2026-01-26

Reject